# Physical Activity and Its Barriers and Facilitators among University Students in Qatar: A Cross-Sectional Study

**DOI:** 10.3390/ijerph19127369

**Published:** 2022-06-16

**Authors:** Karima Chaabna, Ravinder Mamtani, Amit Abraham, Patrick Maisonneuve, Albert B. Lowenfels, Sohaila Cheema

**Affiliations:** 1Institute for Population Health, Weill Cornell Medicine-Qatar, Education City, Qatar Foundation, Doha P.O. Box 24144, Qatar; kac2047@qatar-med.cornell.edu (K.C.); ram2026@qatar-med.cornell.edu (R.M.); ama2006@qatar-med.cornell.edu (A.A.); 2Division of Epidemiology and Biostatistics, IEO European Institute of Oncology IRCCS, 20141 Milan, Italy; patrick.maisonneuve@ieo.it; 3Department of Surgery, New York Medical College, Valhalla, New York, NY 10595, USA; al_lowenfels@nymc.edu; 4Department of Family Medicine, New York Medical College, Valhalla, New York, NY 10595, USA

**Keywords:** physical activity, Qatar, Gulf Cooperation Council, Middle East, exercise, sedentary behavior, sex characteristics, young adult

## Abstract

Physical inactivity is a leading risk factor for non-communicable diseases worldwide. This study investigated physical activity (PA) level among university students in Qatar and assessed other lifestyle and demographic factors associated with PA. A cross-sectional study was conducted between February 2017 and February 2018. A self-administered questionnaire, comprising questions from the International Physical Activity Questionnaire and other validated questionnaires, was used to assess PA and other lifestyle behaviors, including sedentary behavior, stress, sleep, dietary habits, and smoking habits. The results were reported according to the STROBE guidelines. A total of 370 students (response rate = 95.6%) were recruited from eight universities via quota sampling. The prevalence of physically active students—as per the World Health Organization’s recommendation for PA of 150–300 min/week—was 64.9% (75.2% in males and 58.3% in females). Females and students >20 years old were half as likely to be active compared with males and younger students. More males enjoyed getting regular exercise (83.6% vs. 67.7%, *p*-value = 0.002). Time spent sitting was similar during weekdays and weekends (mean time ± SD = 480.8 ± 277.7 min/week vs. 492.1 ± 265.0 min/week). Sports facilities and green spaces appear to help increase PA among university students in Qatar. Public health interventions should focus on improving PA-related perception and knowledge among students to further increase PA participation.

## 1. Introduction

Well-established evidence has demonstrated that physical inactivity, together with other modifiable lifestyle risk factors (e.g., unhealthy diet, short sleep duration, and smoking), is a common risk factor for several non-communicable diseases such as cancer, diabetes, cardiovascular disease, and mental conditions [1,2,3]. Physical inactivity is therefore classified as an important contributor to non-communicable diseases worldwide and a global risk to population health [1]. According to the Qatar National Health Strategy in 2018–2022 [4], 69.7% of deaths in Qatar are due to chronic conditions and 70.1% of Qatari adults are overweight or obese. For optimal health benefits, and to be considered physically active, the World Health Organization (WHO) recommends at least 150–300 min of moderate-intensity physical activity (PA) per week for adults, at least 75–150 min of vigorous-intensity PA per week, or an equivalent combination of moderate- and vigorous-intensity PA throughout the week [5]. The WHO’s global target is to reduce physical inactivity by 10% by 2025 and by 15% through 2030 [6].

In the Middle East and North Africa (MENA), a wide range of adult PA prevalence (i.e., prevalence of adults meeting the WHO recommendations for PA) in the adult general population has been observed, from 13.2% in Sudan to 94.9% in Jordan [7]. In Qatar, PA prevalence has been estimated to be 49.3% [7] among adults. Additionally, a higher PA prevalence among males compared to females has been observed for all MENA countries except Lebanon and Jordan [7,8,9]. The most commonly reported barriers to PA in the region were a lack of suitable sports facilities, time, social support, and motivation; gender and cultural norms; being less educated; harsh weather; and a hot climate [10]. For females, the commonly listed PA barriers were a lack of time, cultural barriers, no access to facilities, the absence of exercise partners, and age, amongst others [10]. Factors reported as facilitators to PA were motivation to gain health benefits, losing or maintaining weight, healthy dietary habits, and recreation [10].

The transition from school to college or university is a critical period during which many students start to make their own decisions related to their lifestyle. At the global level, several studies have shown that a substantial proportion of university students report poor health habits such as physical inactivity, smoking, and short sleep duration [11,12,13,14]. Globally, most university students engage in low levels of physical activity, likely due to their academic workload, lack of self-discipline, and poor access to sports facilities [12,14,15]. The underlying theory in this study is the ecological framework of health behavior, which suggests that behavioral change is a complex process, yet can be elicited through a multi-pronged approach targeting intrapersonal, interpersonal, organizational, and socio-community dimensions [16]. Other dimensions, such as the stage of economic development and cultural factors, sometimes referred to as macro-environmental factors, also play a critical role in the adoption of PA. Our hypothesis is that PA levels among college students in Qatar is associated with demographic characteristics (e.g., age and sex), lifestyle factors (e.g., sleep habits and smoking status), and barriers and facilitators. We anticipate PA participation among females to be lower than their male counterparts, consistent with other findings in the region. PA behaviors that are adopted during this period are gradually ingrained and tend to remain unchanged throughout the life course, and may potentially influence the PA behaviors of future generations [12,17]. Data for the region are currently lacking, and the data yielded in this study can provide useful information when comparing with different socio-cultural customs of other high-income countries in Europe and North America.

A recent scoping review highlighted that the available evidence on university student PA levels is limited to the assessment of differences according to sex and age, and recommended that future research should go beyond assessing only these two demographic factors [18]. Therefore, to identify factors associated with PA levels specifically among males and females, the objective of the study was to conduct a sex-specific examination of PA prevalence, barriers, and facilitators among university students in Qatar and a sex-specific assessment of other lifestyle and demographic factors associated with PA in this student population. The study findings will contribute to developing and implementing effective policies and programs which target the identified factors that contribute to physical activity.

## 2. Materials and Methods

### 2.1. Study Design

A cross-sectional study was conducted between February 2017 and February 2018 within Education City (EC), Doha, Qatar. This study was part of a larger project that aimed to assess nutrition, body mass index, physical activity, and other lifestyle habits of university students in EC, Doha, Qatar [19,20]. This study specifically reports the results related to physical activity and sedentary behavior following the Strengthening the Reporting of Observational Studies in Epidemiology (STROBE) guidelines [21] for reporting cross-sectional studies (Appendix A
Table A1).

### 2.2. Study Setting

Education City (EC, Doha, Qatar) encompasses one national and seven international universities and colleges. Students, faculty, staff, their families, as well as individuals that are not affiliated to any campus at EC have access to several fitness and sport facilities for males and females, swimming pools (with specific hours for families, females only, and males only), an equine center, a golf course, a park, and biking and walking trails in EC.

### 2.3. Primary and Secondary Outcomes

The study’s primary outcome was the prevalence of students whose physical activity levels meet the WHO recommendations (PA prevalence). The study’s secondary outcomes were sedentary behavior and PA barriers and facilitators reported by the students.

### 2.4. Participants, Sampling Method and Procedure

The quota sampling method was used to recruit the students to ensure that the study’s sample population was representative of the EC population with regards to nationality, sex, and the university of study. A study sample size of 370 was calculated based on the estimated total number of EC students at 2741, using a margin of error of 5% and 95% confidence interval levels, and accounting for 11% attrition and incomplete responses. EC students were divided into subgroups based on the university they were enrolled in, their nationality (Qatari/non-Qatari), and sex. The proportions in which these subgroups exist in the EC population were calculated. The study sample of students was recruited while maintaining the same proportions of the EC student subgroups. Therefore, the distribution of students according to their university of enrollment, nationality, and sex was representative of the target population.

Adult students (18 and above) attending university or college at EC were eligible if they understood English and were not pregnant at the time of recruitment. Students were invited to voluntarily participate in the study. There was no form of payment or compensation used as an incentive for recruitment. Participants were approached individually by one of seven trained Weill Cornell Medicine-Qatar students or research personnel. After giving informed consent, participants completed a paper-based, self-administered questionnaire. On average, the questionnaire took about 15 min to complete. Upon completion of the questionnaire, participant weight and height were measured using standardized, portable weighing scales and stadiometers. Following this, participation in the study was considered complete, as there were no follow-up encounters.

### 2.5. Patient and Public Involvement

No patients were involved as participants.

### 2.6. Variables and Measurements

Participants’ sociodemographic characteristics, PA and its barriers and facilitators, sedentary behaviors, dietary patterns, smoking status, sleeping habits, and stress exposure were collected using an anonymized, self-administered, English language questionnaire. The questionnaire utilized questions from validated instruments (reported below) and previously published literature [22,23,24,25,26,27]. The questionnaire was piloted prior to the initiation of recruitment, and some questions were modified as appropriate based on feedback.

The PA level during the past month among students was estimated with the International Physical Activity Questionnaire–Short Form (IPAQ–SF), which is used internationally to obtain comparable PA estimates [28]. The PA level for each student was estimated by calculating the total amount of energy expended for PA in metabolic equivalents per minute (MET·min^−1^). Energy expended for walking is 3.3 MET·min^−1^, moderate physical activity is 4 MET·min^−1^, and vigorous physical activity is 8 MET·min^−1^. A student with a weekly energy expenditure of at least 600 MET·min^−1^ from the total PA consisting of any combination of walking, moderate intensity (e.g., bicycling–light effort, recreational badminton, or vacuuming), and/or vigorous intensity activities (e.g., swimming, football, or jumping rope), was considered physically active as per the WHO guidelines.

Sleep habits were measured with questions from the Pittsburgh Sleep Quality Index [24] covering sleep duration (<7 h vs. ≥7 h) and habitual sleep efficiency (<65%, 65–74%, 75–84%, or ≥85%) during the past month. Habitual sleep efficiency is the ratio of the number of hours slept over the number of hours spent in bed.

Stress levels were measured with the short version of the Perceived Stress Scale 4 (PSS-4) [29]. The PSS-4 score (0 to 16) calculation is based on four questions that ask how often during the last month students “felt unable to control important things in life”, “felt confident about her/his ability to handle personal problems”, “felt things are going her/his way” and “felt difficulties piling up so high that she/he could not overcome them”.

Students were asked to score their overall satisfaction with life by asking how satisfied they felt “these days”, on a scale from zero (not at all satisfied) to ten (completely satisfied).

### 2.7. Statistical Analysis

The analysis was performed on de-identified information. A descriptive analysis was performed to characterize the study population. The distribution of student PA levels according to the categorical variables was assessed using 2-sided Pearson’s chi-squared test. The continuous and categorical variables’ association with PA levels (physically active vs. physically inactive) was assessed using univariate binary logistic regression (Model 0). Significant associations between PA levels and the studied factors identified in the univariate analysis were further investigated using multivariable analysis. Three binary logistic regression models were built to explore collinearity. Model 1 adjusted for age and sex, Model 2 adjusted for all the variables significant in the univariable analysis except “enjoy getting regular exercise”, and “importance of PA every day”, and Model 3 adjusted for all statistically significant confounders in the univariate analysis. Collinearity was explored between the sex of the participants, “enjoy getting regular exercise”, and “importance of PA every day”.

Data analysis was conducted using the SPSS software (IBM SPSS Statistics 27, New York, NY, USA). All statistical tests were two-sided and *p*-values <0.05 were considered statistically significant. As missing data were less than 5%, the omission approach (invalid data were discarded but reported in the result tables as per the STROBE guidelines [21]) was chosen to handle missing data [30].

## 3. Results

### 3.1. Participant Selection and Characteristics

Out of the 387 students invited to participate in the study, 370 students (response rate = 95.6%; 61.8% females; mean age ± standard deviation (SD) = 20.1 ± 3.0; age range = 18–39) completed the survey. The reasons for refusal to participate in the study included a lack of time and being self-conscious about personal weight. Students enrolled in this study were from 44 nationalities from the five continents. The included participants’ characteristics are reported in Table 1. Significant differences between study population groups were representative of the differences observed in the EC population.

### 3.2. Time Spent in Physical Activity and Sedentary Behaviors

The time reported for PA per week varied according to PA intensity (vigorous intensity: mean time ± SD = 39.0 ± 49.2 min/week; moderate intensity: mean time ± SD = 36.7 ± 59.3 min/week; and walking time: mean time ± SD = 94.2 ± 144.4 min/week).

The time reported for sitting was similar during weekdays and weekends (mean time ± SD = 480.8 ± 277.7 min/week vs. 492.1 ± 265.0 min/week, respectively). On average, students spent 75.3 ± 105.0 min/week watching television and 417.6 ± 280.3 min/week using either a computer, laptop, or phone.

### 3.3. Participants’ Characteristics and Physical Activity

The prevalence of PA was 64.9% (240/370) among the EC students. Age and sex were significantly associated factors with PA in both the univariable and multivariable analyses (Table 2). Students under 20 years old were significantly more likely to be physically active than students 20 years old and over. This association became even more significant when other confounders were included in the model (Models 2 and 3).

Female students were about 50% less likely to be physically active than male students (statistically significant in Models 0, 1, and 2). However, this association of PA levels with sex disappeared when an adjustment for all the significant predictors at univariable analysis was considered. An examination of collinearity demonstrated a correlation between sex, enjoyment of getting regular exercise, and the perceived importance of PA every day (Table 3). Significantly more males enjoyed getting regular exercise (83.6% vs. 67.7%, *p*-value = 0.002) and reported that PA is extremely important (54.6% vs. 28.1%, *p*-value < 0.001).

The other studied participants’ characteristics (i.e., country of origin, field of study at university, education program, place of residence—on-campus or off-campus, and nationality—Qatari vs. non-Qatari) were not statistically associated with PA levels in EC students.

### 3.4. Mental Health and Physical Activity

The reported smoking status, perceived stress as measured with the PSS-4 tool, sleep duration and habitual sleep efficacy as measured by the Pittsburgh Sleep Quality Index [29], concentration, and feeling of satisfaction with life as a whole were not statistically associated with PA levels. The results did not change when the assessment of PA was conducted separately for male and female students (Table 4).

### 3.5. Dietary Patterns and Physical Activity

An increased frequency in eating breakfast and lunch (per week) were statistically associated with an increased likelihood of meeting the WHO’s PA level guidelines (in univariable and multivariable analyses, Model 1) (Table 2). Conversely, an increased frequency of eating while watching television (per week) was statistically associated with a decreased likelihood of meeting the WHO’s PA level guidelines (in univariable and multivariable analyses, Model 1). However, after adjustment with Models 2 and 3, associations with an increased frequency of eating breakfast and lunch and eating while watching television were not statistically significant. Other eating patterns, such as eating dinner, eating at late hours of the night, and eating between meals, were not associated with PA levels.

Reporting being too busy to eat healthy food was significantly associated with lower PA levels in univariable and multivariable analyses (Model 1), but this association disappeared in Models 2 and 3. The perceived importance of eating healthy foods and the perceived costliness of healthy foods were not associated with PA levels among males and females (Table 5).

Attempting to lose weight and the self-perceived weight (i.e., perceived weight as underweight, normal, overweight, or obese) were not associated with PA levels among males and females. Additionally, emotional eating, e.g., eating when bored, stressed, or angry, was also not associated with PA levels. Being a vegetarian, taking any types of supplements, and consuming protein shakes and energy drinks were also not associated with PA levels.

### 3.6. PA Barriers and Facilitators

Seventy percent of the students reported a lack of time as a barrier to regular exercise (Table 6). However, no statistical association was identified between a lack of time and PA levels. Most of the students did not identify the following barriers as applying to them: being self-conscious and concerned with sociocultural norms that prevent them from exercising (95%), a lack of facilities and space (92%), environmental conditions (e.g., temperature and safety; 90%), a lack of interest (87%), a lack of knowledge on how to exercise (86%), a lack of an exercise buddy (80%), and a lack of self-discipline (70%).

A total of 72% of the students who reported enjoying getting regular exercise were physically active and 78% of the students who reported that it is extremely important to do PA every day were physically active.

Enjoying regular PA and the perceived importance of daily PA were statistically associated with a higher level of PA among students (in univariable and multivariable analyses). However, while the associations between PA levels and these two facilitators were statistically significant in univariable and multivariable analyses (Model 1), these associations disappeared after adjustment with the full model (Model 3). Stratified analysis (Table 6) showed that both male and female physically active students enjoyed PA more than inactive students, and males enjoyed PA more than females. Additionally, both male and female physically inactive students agreed less frequently with the statement affirming the extreme importance of PA every day. However, while most physically active male students agreed with the extreme importance of PA, most of their female counterparts reported that PA is “somewhat important”.

## 4. Discussion

In summary, PA prevalence among students in EC was estimated at 64.9% in 2018 (males: 75.2% and females: 58.3%). While several factors (sociodemographic, lifestyle, barriers, and facilitators) were assessed, the only statistically significant factors associated with PA level among these students were age and sex. More specifically, PA levels appeared to be higher among students younger than 20 years old and 50% lower among females.

In Qatar, a previous meta-analysis estimated the prevalence of PA to be 49.3% among adults over the period of 2003–2015 [7]. This study therefore indicates that the PA prevalence among EC students is higher than that estimated for adults in Qatar a few years ago. Despite a dominant car-centric culture in Qatar, substantial national efforts have been made to enhance PA levels in the country. Worldwide, Qatar is one of the few countries with a dedicated national sports day [31]. In recent years, many recreational resources such as green parks, biking and walking trails, outdoor fitness machines, and open spaces have been made accessible free of charge. As Qatar has very hot weather for 6–8 months of the year, a “mall walking program” was launched to allow walking indoors on hot days [32]. This represents a problem-based solution to the difficulties of undertaking physical activity in a challenging environment. Community environmental considerations such as aesthetics, cleanliness, and safety have been also addressed in Qatar [33]. For instance, walkable neighborhoods with sidewalks, streetlights, public transport, shops, and restaurants have been developed. Additionally, Qatar is one of the safest countries in the world. A well-designed environment can contribute to increasing PA, preventing premature mortality, and reducing costs associated with physical inactivity [34,35,36]. Our study might reflect the increase in PA among Qatar’s adults occurring before the COVID-19 pandemic when many construction projects were delivered.

While the built environment is crucial for promoting PA in a population, individual and social factors are also important [37]. Our study population was diverse, including students from 44 nationalities from all over the world. This diversity likely reflects Qatar society, which has a unique demographic, since non-Qataris come from over 150 countries and encompass 89% of the population [38,39]. No differences in PA levels between nationalities (Qataris/non-Qataris), region of origin, place of residence (on-campus/off-campus), or field of study at the university were identified. EC students from all origins seemed to participate similarly in PA.

More males were physically active than females, as previously observed worldwide [40]. After adjusting for all confounders, significantly lower PA levels were no longer detectable among females. This lack of significance could be due to an improvement in PA levels in female students and/or due to statistical matters related to correlated independent variables included in the multivariable analysis. However, when some confounders were dropped from the full model due to collinearity, female students were 50% significantly less active than male students. Collinearity was identified between the sex of participants, “enjoy getting regular exercise”, and “importance of PA every day”. Compared with their male counterparts, female students reported enjoying exercise less and were less likely to describe PA as extremely important. Therefore, differences in the PA-related perception and knowledge between male and female students is likely one of the explanations for PA level differences between both sexes. Consequently, efforts to improve the perception and knowledge about PA should be included in community and behavioral public health interventions to further increase PA participation among female students. Programs such as the This Girl Can campaign in England [41], which aims to improve knowledge and perception among females, may further increase PA participation among females after being culturally adapted [40].

Additionally, barriers previously reported among adults [10], such as self-consciousness and sociocultural norms that prevent some people from exercising, were not identified as barriers by over 90% of EC students. Our findings suggest that potential sex-based inequality is likely being addressed by the offering of a large variety of culturally adapted options for PA, such as swimming pools with hours for women only, men only, and families, and separate gyms for males and females. While there was likely an increase in PA levels among female students as a result of them using the new facilities available in EC, it appears from our findings that there are still differences in PA levels among male and female students. An assessment of the impact of addressing these barriers requires longitudinal data, which are not available.

Our findings have demonstrated that EC students younger than 20 years old are twice as likely to be active as older students. While entering EC, students are offered a large choice of sport activities and facilities free of charge, which might be new to them and represent a good way for them to socialize. However, during the following years, students have likely developed established networks of friends and may have to cope with an increased workload, resulting in less time for and/or prioritizing of sport activities. Course load has been shown to be negatively associated with vigorous PA and positively associated with sedentary behavior [42]. Our findings emphasize the need for the development of campus-based health initiatives to increase PA among the older students.

The objective of our study was to investigate several lifestyle habits of students from eight universities in Doha. The quota sampling method was used to recruit the students. Therefore, the distribution of students according to their university, nationality, and sex was representative of the target population. A high response rate (95%) to the questionnaire was obtained from the EC students through the use of short-form instruments such as IPAQ and PSS-4 rather than the more comprehensive instruments. Satisfaction with life was also assessed using a single Likert scale.

As the study was cross-sectional, different results might have been obtained if the study had been conducted at a different time of the year, such as during vacations or the exam period. Education City is a unique and diverse educational environment in the MENA region. However, it is likely that our findings are generalizable to the students attending similar diverse campuses in the high-income countries of the region (e.g., the United Arab Emirates or Kuwait) and other countries with diverse student demographics (e.g., the United States of America or the United Kingdom). In most primary studies pertaining to the MENA high-income countries, where non-nationals represent over 35% of the population, the distinction between the national and non-national populations are unclear [7]. Our investigation of PA levels by nationality demonstrated that students have similar PA levels regardless of their origin.

## 5. Conclusions

Overall, while several factors (sociodemographic, lifestyle, barriers, and facilitators) were assessed, the only statistically significant factors associated with PA level among these college students were age and sex. Older and female students were less active than their younger and male counterparts. PA promotion constitutes an opportunity to address several population health priorities and to help achieve the United Nations General Assembly’s Sustainable Development Goals [43]. The implementation of COVID-19 safety measures (e.g., lockdowns and fitness club closures) has been a challenge to promoting PA. Nevertheless, promoting PA in everyday life can leverage multiple benefits for physical and mental health that is critical during the COVID-19 pandemic. Our findings demonstrate that just before the pandemic, nearly two-thirds of the students in EC, Qatar were physically active as recommended by the WHO. However, female students and students older than 20 years were statistically less active. Fewer female students reported enjoying exercise compared to male students and more male students considered PA every day to be extremely important. The lower female perception of—and knowledge about—PA identified in our study explains the male–female differences related to physical activity. Efforts to improve the perception and knowledge about PA should be included in community and behavioral public health interventions to further increase PA participation among female students. The FIFA World Cup 2022™ is an opportune moment for Qatar to revive young adult hearts and further promote PA.

## Figures and Tables

**Table 1 ijerph-19-07369-t001:** Participant characteristics according to their physical activity levels.

	All Participants		Male				Χ2 Test *p*-Value	Female				Χ2 Test *p*-Value
			Inactive		Active			Inactive		Active		
	N	(%)	N	(%)	N	(%)		N	(%)	N	(%)	
**All**	370	(100)	35	(24.8)	106	75.2		95	(41.3)	133	(58.3)	
**Age**												
<20 years old	192	(100)	10	(14.5)	59	(85.5)	0.005	45	(36.6)	78	(63.4)	0.112
≥20 years old	174	(100)	25	(34.7)	47	(65.3)		48	(47.1)	54	(52.9)	
Missing data	4	(1.1)	-	-	-	-		-	-	-	-	
**Country of Origin**												
Asia, Africa	36	(100)	5	(27.8)	13	(72.2)	0.160	11	(55.6)	7	(38.9)	0.246
Europe, America, Oceania	42	(100)	1	(5.3)	18	(94.7)		7	(81)	16	(69.6)	
EMR	110	(100)	17	(31.5)	37	(68.5)		22	(64.9)	34	(60.7)	
Qatar	163	(100)	11	(26.8)	30	(73.2)		52	(61.3)	70	(57.4)	
Missing data	19	(5.1)	-	-	-	-		-	-	-	-	
**Field of Study at University**												
Fine Arts	48	(100)	1	(33.3)	2	(66.7)	0.012	23	(51.1)	22	(48.9)	0.119
Medicine	41	(100)	4	(19.0)	17	(81.0)		10	(50.0)	10	(50.0)	
Engineering	87	(100)	6	(12.5)	42	(87.5)		18	(46.2)	21	(53.8)	
Business/computer studies	80	(100)	17	(44.7)	21	(55.3)		14	(33.3)	28	(66.7)	
Foreign service	37	(100)	3	(27.3)	8	(72.7)		6	(23.1)	20	(76.9)	
Journalism	31	(100)	2	(22.2)	7	(77.8)		6	(27.3)	16	(72.7)	
Archeology	14	(100)	2	(66.7)	1	(33.3)		5	(45.5)	6	(54.5)	
Academic Bridge Program	31	(100)	0	(0)	8	(100)		13	(56.5)	10	(43.5)	
Missing data	1	(0.3)	-	-	-	-		-	-	-	-	
**Education Program**												
Undergraduate	340	(100)	28	(21.5)	102	(78.5)	0.002	89	(42.4)	121	(57.6)	0.455
Graduate	29	(100)	7	(63.6)	4	(36.4)		6	(33.3)	12	(66.7)	
Missing data	1	(0.3)	-	-	-	-	-	-	-	-	-	
**Place of Residence**												
On-campus	93	(100)	17	(32.7)	35	(67.3)	0.74	17	(41.5)	24	(58.5)	0.74
Off-campus	276	(100)	18	(20.2)	71	(79.8)		78	(41.7)	109	(58.3)	
Missing data	1	(0.5)	-	-				-	-			
**Nationality**												
Non-Qatari	205	(100)	24	(24.0)	76	(76.0)	0.186	42	(40.0)	63	(60.0)	0.186
Qatari	163	(100)	11	(26.8)	30	(73.2)		52	(42.6)	70	(57.4)	
Missing data	2	(0.3)	-	-	-	-		-	-			

**Table 2 ijerph-19-07369-t002:** Assessment of the factors associated with increased physical activity.

	Model 0		Model 1		Model 2		Model 3	
	Unadjusted OR	95% CI	Adjusted OR	95% CI	Adjusted OR	95% CI	Adjusted OR	95% CI
**Age**								
≥20 years old	Ref.	-	Ref.	-	Ref.	-	Ref.	-
<20 years old	**1.81**	**1.17–2.8**	**1.92**	**1.23–2.99**	**2.08**	**1.30–3.33**	**2.15**	**1.32–3.50**
**Sex**								
Male	Ref.	-	Ref.	-	Ref.	-	Ref.	-
Female	**0.46**	**0.29–0.73**	**0.44**	**0.28–0.71**	**0.49**	**0.29–0.81**	0.61	0.36–1.04
**Enjoy getting regular exercise**								
Agree/strongly agree	Ref.	**-**	Ref.	**-**	-	-	Ref.	-
Neither agree nor disagree	**0.39**	**0.23–0.67**	**0.42**	**0.24–0.73**	-	-	0.57	0.31–1.06
Disagree/strongly disagree	**0.30**	**0.13–0.71**	**0.32**	**0.14–0.75**	-	-	0.54	0.20–1.43
**Importance of physical activity every day**								
Not important	Ref.	**-**	Ref.	**-**	-	-	Ref.	-
Somewhat important	1.31	0.65–2.61	1.32	0.65–2.67	-	-	0.86	0.38–1.92
Extremely important	**3.37**	**1.60–7.09**	**3.10**	**1.45–6.64**	**-**	**-**	1.68	0.68–4.16
**Being too busy to eat healthy foods**								
Agree/Strongly agree	Ref.	-	Ref.	-	Ref.	-	Ref.	-
Neither agree nor disagree	1.41	0.85–2.36	1.36	0.80–2.31	1.24	0.71–2.17	1.20	0.68–2.14
Disagree/Strongly disagree	**2.19**	**1.23–3.89**	**2.03**	**1.12–3.66**	1.59	0.85–2.99	1.45	0.75–2.80
**Protein shake consumption ****	**1.18**	**1.00–1.39**	1.14	0.95–1.35	0.93	0.80–1.08	0.95	0.81–1.12
**Eat breakfast *****	**1.26**	**1.03–1.54**	**1.31**	**1.07–1.61**	1.17	0.93–1.47	1.06	0.83–1.35
**Eat lunch *****	**1.51**	**1.13–2.03**	**1.50**	**1.11–2.04**	1.39	0.99–1.95	1.34	0.94–1.91
**Eat while watching TV *****	**0.81**	**0.67–0.99**	**0.81**	**0.66–0.99**	0.81	0.66–1.00	0.83	0.67–1.03

Model 0: univariate binary logistic regression; Model 1: adjustment for sex and age; Model 2: adjustment for sex, age, being too busy to eat healthy food, eating breakfast, eating lunch, and eating while watching TV; and Model 3: adjustment for all variables. Bold values of unadjusted OR and adjusted OR: Wald test ≤0.05. ** 1: never or less than one per month; 2: 1–3 per month; 3: once a week; 4: 2–4 per week; 5: 5–6 per week; 6: once a day; 7: 2–3 per day; 8: 4–5 per day; and 9: 6+ per day. *** 1: never; 2: 1–3 times/week; 3: 4–6 times/week; and 4: daily. Bold numbers mean that the statistical test was significant for the corresponding category.

**Table 3 ijerph-19-07369-t003:** Exploring collinearity between sex and other independent variables.

	All Participants		Males		Females		Pearson Χ2 Test *p*-Value
	N	(%)	N	(%)	N	(%)	
**All**	370	(100)	147	(38.6)	223	(61.4)	
**Enjoy getting regular exercise**							
Disagree/strongly disagree	25	(100)	8	(32.0)	17	(68.0)	**0.002**
Neither agree nor disagree	71	(100)	15	(21.1)	56	(78.9)	
Agree/strongly agree	267	(100)	117	(43.8)	150	(56.2)	
Missing data	6	(1.9)	-	-	-	-	
**Importance of physical activity every day**							
Not important	39	(100)	14	(35.9)	25	(64.1)	**>0.001**
Somewhat important	190	(100)	50	(26.5)	139	(73.5)	
Extremely important	141	(100)	77	(54.6)	64	(45.4)	
Missing data	0	(0)	-	-	-	-	

Bold numbers mean that the statistical test was significant for the corresponding category.

**Table 4 ijerph-19-07369-t004:** Physical activity and mental-health-related factors.

	All Participants		Male				Χ2 Test *p*-Value	Female				Χ2 Test *p*-Value
			Inactive		Active			Inactive		Active		
	N	(%)	N	(%)	N	(%)		N	(%)	N	(%)	
**Perceived stress (PSS-4 scale)**												
Not feeling stressed (score 0–7)	190	(100)	27	(30.0)	63	(70.0)	0.067	35	(35.0)	65	(65.0)	0.128
Feeling stressed (score 8–16)	172	(100)	8	(16.0)	42	(84.0)		55	(45.1)	67	(54.9)	
Missing data	8	(2.2)	-	-	-	-		-	-	-	-	
**Sleep duration**												
<7 h	211	(100)	20	(23.0)	67	(77.0)	0.647	50	(40.3)	74	(59.7)	0.869
≥7 h	152	(100)	14	(26.4)	39	(73.6)		41	(41.1)	58	58.6)	
Missing data	7	(1.9)	-	-	-	-		-	-	-	-	
**Habitual sleep efficiency**												
≥85%	18	(100)	1	(20.0)	4	(80.0)	0.401	8	(61.5)	5	(38.5)	0.299
75–84%	28	(100)	2	(16.7)	10	(83.3)		7	(43.8)	9	(56.3)	
65–74%	74	(100)	4	(14.3)	24	(85.7)		15	(32.6)	31	(67.4)	
<65%	241	(100)	27	(28.7)	67	(71.3)		61	(41.5)	86	(58.5)	
Missing data	9	(2.4)	-	-	-	-		-	-	-	-	
**Concentration**												
No	147	(100)	13	(21.0)	49	(79.0)	0.414	35	(41.2)	50	(58.8)	0.98
Yes	217	(100)	21	(26.9)	57	(73.1)		57	(41.0)	82	(59.0)	
Missing data	6	(1.6)	-	-	-	-		-	-	-	-	
**Smoker**												
Never or once	299	(100)	24	(22.4)	83	(77.6)	0.206	82	(42.7)	110	(57.3)	0.053
Smoker	56	(100)	11	(33.3)	22	(66.7)		5	(21.7)	18	(78.3)	
Missing data	15	(4.1)	-	-	-	-		-	-	-	-	

**Table 5 ijerph-19-07369-t005:** Physical activity and dietary-related factors.

	All Participants		Male				Χ2 Test *p*-Value	Female				Χ2 Test *p*-Value
			Inactive		Active			Inactive		Active		
	N	(%)	N	(%)	N	(%)		N	(%)	N	(%)	
**Vegetarian**												
No	350	(100)	34	(25.2)	101	(74.8)	0.636	93	(43.3)	122	(56.7)	0.48
Yes	19	(100)	1	(16.7)	5	(83.3)		2	(15.4)	11	(84.6)	
Missing data	1	(0.3)	-	-	-	-		-	-	-	-	
**Supplements**												
No	262	(100)	30	(26.5)	83	(73.5)	0.232	63	(42.3)	86	(57.7)	0.796
Yes	105	(100)	4	(15.4)	2	(84.6)		32	(40.5)	47	(59.5)	
Missing data	3	(0.8)	-	-	-	-		-	-	-	-	
**Self-perceived weight**												
Underweight	42	(100)	1	(6.3)	15	(93.8)	0.061	11	(42.3)	15	(57.7)	0.631
Normal	209	(100)	17	(21.8)	61	(78.2)		58	(44.3)	73	(55.7)	
Overweight	108	(100)	16	(38.1)	26	(61.9)		25	(37.9)	41	(62.1)	
Obese	10	(100)	1	(20.0)	4	(80.0)		1	(20.0)	4	(80.0)	
Missing data	1	(0.3)	-	-	-	-		-	-	-	-	
**Weight loss attempt**												
No	179	(100)	17	(21.5)	62	(78.5)	0.305	45	(45.0)	55	(55.0)	0.393
Yes	189	(100)	18	(29.0)	44	(71.0)		50	(39.4)	77	(60.6)	
Missing data	2	(0.5)	-	-	-	-		-	-	-	-	
**Importance for him/her to eat healthy foods**												
Disagree/Strongly disagree	21	(100)	1	(11.1)	8	(88.9)	0.123	5	(41.7)	7	(58.3)	0.562
Neither agree or disagree	47	(100)	6	(46.2)	7	(53.8)		17	(50.0)	17	(50.0)	
Agree/Strongly agree	301	(100)	28	(23.5)	91	(76.5)		73	(40.1)	109	(59.9)	
Missing data	1	(0.3)	-	-	-	-		-	-	-	-	
**Being too busy to eat healthy foods**												
Disagree/Strongly disagree	85	(100)	7	(17.9)	32	(82.1)	0.131	14	(30.4)	32	(69.6)	0.202
Neither agree or disagree	99	(100)	8	(19.0)	34	(81.0)		25	(43.9)	32	(56.1)	
Agree/Strongly agree	181	(100)	20	(33.3)	40	(66.7)		55	(45.5)	66	(54.5)	
Missing data	5	(1.4)	-	-	-	-		-	-	-	-	
**Perceiving healthy food as too expensive**												
Disagree/Strongly disagree	114	(100)	5	(12.8)	34	(87.2)	0.107	31	(41.3)	44	(57.1)	0.964
Neither agree or disagree	108	(100)	10	(26.3)	28	(73.7)		30	(42.9)	40	(59.3)	
Agree/Strongly agree	145	(100)	20	(31.3)	44	(68.8)		33	(40.7)	48	(58.4)	
Missing data	3	(0.8)	-	-	-	-		-	-	-	-	

**Table 6 ijerph-19-07369-t006:** Barriers and positive perceptions towards physical activity.

	All Participants		Male				Χ2 Test *p*-Value	Female				Χ2 Test *p*-Value
			Inactive		Active			Inactive		Active		
	N	(%)	N	(%)	N	(%)		N	(%)	N	(%)	
**Lack of time**												
No	109	(100)	14	(25.5)	41	(74.5)	0.841	20	(37.0)	34	(63.0)	0.527
Yes	257	(100)	20	(23.5)	65	(76.5)		74	(43.0)	98	(57.0)	
Missing data	4	(1.1)	-	-	-	-	-	-	-	-	-	-
**Not interested**												
No	319	(100)	28	(22.5)	95	(77.2)	0.364	78	(39.8)	118	(60.2)	0.170
Yes	47	(100)	6	(35.5)	11	(64.7)		16	(53.3)	14	(46.7)	
Missing data	4	(1.1)	-	-	-	-	-	-	-	-	-	-
**Lack of self-discipline**												
No	252	(100)	21	(21.0)	79	(79.0)	0.191	64	(42.1)	88	(57.9)	0.886
Yes	114	(100)	13	(32.5)	27	(67.5)		30	(40.5)	44	(59.5)	
Missing data	4	(1.1)	-	-	-	-	-	-	-	-	-	-
**Lack of facilities and space**												
No	335	(100)	31	(23.3)	102	(76.7)	0.360	83	(43.1)	119	(58.9)	0.667
Yes	31	(100)	3	(42.9)	4	(57.1)		11	(45.8)	13	(54.2)	
Missing data	4	(1.1)	-	-	-	-	-	-	-	-	-	-
**Self-conscious/sociocultural norms preventing exercise**												
No	346	(100)	33	(24.1)	104	(75.9)	0.569	88	(42.1)	121	(57.9)	0.621
Yes	20	(100)	1	(33.3)	2	(66.7)		6	(35.3)	11	(67.4)	
Missing data	4	(1.1)	-	-	-	-	-	-	-	-	-	-
**Environmental conditions (temperature/safety)**												
No	329	(100)	33	(24.6)	101	(75.4)	1.000	78	(40.0)	117	(60.0)	0.243
Yes	37	(100)	1	(16.7)	5	(83.3)		16	(51.6)	15	(48.4)	
Missing Data	5	(1.4)	-	-	-	-	-	-	-	-	-	-
**Lack of knowledge on how to exercise**												
No	314	(100)	28	(22.6)	96	(77.4)	0.218	78	(41.1)	112	(58.9)	0.854
Yes	51	(100)	6	(37.5)	10	(62.5)		15	(42.9)	20	(57.1)	
Missing data	4	(1.1)	-	-	-	-	-	-	-	-	-	-
**Lack of an exercise buddy**												
No	292	(100)	26	(22.4)	90	(70.6)	0.297	69	(39.2)	107	(60.8)	0.195
Yes	74	(100)	8	(33.3)	16	(66.7)		25	(50.0)	25	(50.0)	
Missing data	4	(0.3)	-	-	-	-	-	-	-	-	-	-
**Importance of physical activity every day**												
Disagree/strongly disagree	39	(100)	5	(35.7)	9	(64.3)	0.056	14	(56.0)	11	(44.0)	**0.021**
Neither agree nor disagree	189	(100)	17	(34.0)	33	(66.0)		63	(45.3)	76	(54.7)	
Agree/strongly agree	141	(100)	13	(16.9)	64	(83.1)		18	(28.1)	46	(71.9)	
Missing data	1	(0.3)	-	-	-	-	-	-	-	-	-	-
**Enjoying getting regular exercise**												
Not important	25	(100)	6	(75.0)	2	(25.0)	**0.001**	8	(47.1)	9	(52.9)	**0.023**
Somewhat important	71	(100)	5	(33.3)	10	(66.7)		31	(55.4)	25	44.6	
Extremely important	267	(100)	23	(19.7)	94	(80.3)		52	(34.7)	98	(65.3)	
Missing data	1	(0.3)	-	-	-	-	-	-	-	-	-	-

Bold numbers mean that the statistical test was significant for the corresponding category.

## Data Availability

The data that support the findings of this study are available on request from the corresponding author, S.C. The data are not publicly available due to ethical restrictions; they contain information that could compromise the privacy of research participants.

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
