# Peer review of "Physical Activity and Its Barriers and Facilitators among University Students in Qatar: A Cross-Sectional Study"

_ijerph, 2022, doi:10.3390/ijerph19127369_

Round 1

Reviewer 1 Report

This study aimed to conduct a comprehensive assessment of Physical Activity and its Barriers, and Facilitators among University Students in Qatar.

This contribution seems to be relevant and interesting considering the aim of the article and the approach used. That has been poorly addressed in the literature in this geogrphical area. However, there are limitations or aspects that need improvement that must be corrected before the publication in the journal.

The authors have made a good effort in trying to carry out their study according to a scientific rationale and with an appropriate methodology. In my opinion, the results concerning the differences between groups are not novel and have been previously reported in the literature in other populations, but there is an unbalance between the different groups included in this study

Abstract: Description of Methods are uncomplete. Conclusion should explain the motivation of results and practical application. There isn’t a take home message.

Introduction: The introduction does not introduce correctly the background and state of art. First of all the literature review and the theoretical framework seems to be weak, considering the extent of the introduction and the rest of the article this should be expanded including a thorough review of the literature and exposing the major limitations of other studies emphasizing the contributions this work represents compared to previous work. Currently, there are a lot of manuscripts about physical activity and main barriers in students. Secondly, it should deepen on the interest and justification this article represents not only to the journal but also to the discipline as a whole.

Ultimately it is necessary to increase the "sale" arguments of this article for future readers.

Methods: Please, increase the study design. Some sections are uncomplete and they should be rewrite with major precision and accuracy. Considering participants section necessary information on the composition of the sample it is not included. When exposing the “measures” and the “procedure” the authors don’t include necessary information as: • Duration of the surveys, • Number of interviewers or observers in each survey, • If it was always the same interviewer or were more than one.

Results: With regard to the analysis and results reported. The article would be enriched if the results were explained better. I cannot see the effect size or the confidence interval of the differences. The statistical analysis must be improved and the groups should be more balanced.

Discussion: Authors should give readers a "take-home" message and you should improve the practical application in this field.

References: There are format and/or spelling mistake. Please revise

Author Response

Response: We would like to thank the reviewer for the thoughtful comments. As per the reviewer’s suggestions we have now revised the manuscript. Kindly find below our response to the specific comments.

Comments and Suggestions for Authors

Comment: This study aimed to conduct a comprehensive assessment of Physical Activity and its Barriers, and Facilitators among University Students in Qatar. This contribution seems to be relevant and interesting considering the aim of the article and the approach used. That has been poorly addressed in the literature in this geogrphical area.

Response: We would like to thank the reviewer for appreciating the relevance of this study.

Comment: However, there are limitations or aspects that need improvement that must be corrected before the publication in the journal.

The authors have made a good effort in trying to carry out their study according to a scientific rationale and with an appropriate methodology. In my opinion, the results concerning the differences between groups are not novel and have been previously reported in the literature in other populations, but there is an unbalance between the different groups included in this study.

Response: To minimize selection bias, we used quota sampling method to recruit the students. The number of students per sex, nationality, and university of study is representative of the number of students for these categories in the Education City campuses. Therefore, the differences observed between study population groups are representative of the differences observed in the target population.

To clarify this matter more detail has now been provided in the Materials and Methods section and in the ‘Participants selection and characteristics’ subsection of the Result section.

Comment: Abstract: Description of Methods are uncomplete. Conclusion should explain the motivation of results and practical application. There isn’t a take home message.

Response: Thank you for the suggestion. We have now added to the methods and conclusion as much as possible while keeping within the word limit of 200 words for the abstract as indicated in the IJERPH’s author guidelines.

Comment: Introduction: The introduction does not introduce correctly the background and state of art. First of all the literature review and the theoretical framework seems to be weak, considering the extent of the introduction and the rest of the article this should be expanded including a thorough review of the literature and exposing the major limitations of other studies emphasizing the contributions this work represents compared to previous work. Currently, there are a lot of manuscripts about physical activity and main barriers in students. Secondly, it should deepen on the interest and justification this article represents not only to the journal but also to the discipline as a whole.

Ultimately it is necessary to increase the "sale" arguments of this article for future readers.

Response: We thank the reviewer for the thoughtful feedback. As suggested by the reviewer, we have now introduced a theoretical framework and expanded the introduction by adding more references to deepen the interest and justify the publication of this article.

Comment: Methods: Please, increase the study design. Some sections are uncomplete and they should be rewrite with major precision and accuracy. Considering participants section necessary information on the composition of the sample it is not included. When exposing the “measures” and the “procedure” the authors don’t include necessary information as: • Duration of the surveys, • Number of interviewers or observers in each survey, • If it was always the same interviewer or were more than one.

Response: As per the reviewer’s suggestion, we have provided additional information in the Materials and Methods section. ‘Participants’ subsection and ‘Sampling’ subsection are now more detailed and have been merged. Detailed description of the composition of the sample are provided in ‘Participants selection and characteristics’ subsection of the Result section and in Table 1.

Comment: Results: With regard to the analysis and results reported. The article would be enriched if the results were explained better. I cannot see the effect size or the confidence interval of the differences. The statistical analysis must be improved and the groups should be more balanced.

Response: We did not calculate differences between groups, rather we assessed the association between PA levels and the selected factors (demographic, lifestyle, barriers, and facilitators), by computing odds ratios (effect size) along with their 95% confidence intervals using unadjusted and adjusted logistic regression model. The rationale for using odds ratios and not differences as effect size is based on the study’s objective that is to compare the prevalence of students meeting the World Health Organization’s recommendation for physical activity and not compare the questionnaire’s mean score for physical activity level between students’ groups.

Stratified statistical analysis has been conducted. Result section and Tables 1, 3, 4, 5 have been updated.

Comment: Discussion: Authors should give readers a "take-home" message and you should improve the practical application in this field.

Response: As per the reviewer’s suggestion, in the conclusion section a “take-home” message has been added and practical application in this field has been improved in the Discussion section.

Comment: References: There are format and/or spelling mistake. Please revise

Response: Formatting and/or spelling errors have been fixed.

Reviewer 2 Report

The introduction can be improved and supported by other studies.

In the same way, the methodology is correct. However, the methodological design should be more descriptive and supported by other authors.

I think the conclusions are correct but it could be deepened more according to the results achieved.

Article is originality with significance of content.

I think it can be published with minor modifications.

Author Response

Response: Many thanks for your feedback on our manuscript. As per the reviewer’s suggestions, the manuscript has been revised. Kindly find below our response to the specific comments.

Comments and Suggestions for Authors

Comment: The introduction can be improved and supported by other studies.

Response: We have improved the introduction supported by additional studies.

Comment: In the same way, the methodology is correct. However, the methodological design should be more descriptive and supported by other authors.

Response: As per the reviewer’s suggestion, the Materials and Methods section is now more descriptive and supported by other authors.

Comment: I think the conclusions are correct but it could be deepened more according to the results achieved.

Response: We have now deepened the conclusions to resonate with the results achieved.

Comment: Article is originality with significance of content.

I think it can be published with minor modifications.

Response: We would like to thank the reviewer for appreciating the relevance of this study and for the positive comments.

Reviewer 3 Report

There is nothing fundamentally wrong with this descriptive study. The research holds together and the findings are logical.  However, I personally believe the most important issue in this paper is the question:  why aren't women participating in exercise and physical activity if all of the support services are available for women.  Second, the authors ran all the data as a total except in the first instance of gender.  The gender difference is the "elephant in the room".  Because there is such a discrepancy in the N sizes of gender, the data may be questionable to compare them - I understand that.  But running everything as a total is also not an honest perspective of what is found.  The good news is the men in education in Qatar are being active and appear to live their lives according to good fundamental exercise principles.  However, maybe not since both genders were aggregated into the tables 1, 3, 4, 5, and the data was discussed by PA or non PA.  I would argue that of course the data would be what the researchers found.  The more important question is about gender and the differences and I wonder about the instruments used to measure the factors.  The barriers for women getting enough PA in general is about them being women - child care, elder parent care, family care, - in other words time available to do so.  So, even though I find nothing fundamentally wrong with this study, I don't think the results or the discussion of the results are/is discussing the real problem for PA in collegiate students in Qatar.  I would like the authors to discuss this topic further and offer us the data by gender on all tables. 

Author Response

Response: Many thanks for your thoughtful feedback. As per the reviewer’s suggestions we have now revised the manuscript. Kindly find below our response to the specific comments.

Comments and Suggestions for Authors

Comment: There is nothing fundamentally wrong with this descriptive study. The research holds together and the findings are logical.

Response: We would like to thank the reviewer for appreciating the relevance of this study.

Comment: However, I personally believe the most important issue in this paper is the question:  why aren't women participating in exercise and physical activity if all of the support services are available for women.

Second, the authors ran all the data as a total except in the first instance of gender. The gender difference is the "elephant in the room". Because there is such a discrepancy in the N sizes of gender, the data may be questionable to compare them - I understand that. But running everything as a total is also not an honest perspective of what is found.  The good news is the men in education in Qatar are being active and appear to live their lives according to good fundamental exercise principles. However, maybe not since both genders were aggregated into the tables 1, 3, 4, 5, and the data was discussed by PA or non PA. I would argue that of course the data would be what the researchers found. The more important question is about gender and the differences and I wonder about the instruments used to measure the factors.

The barriers for women getting enough PA in general is about them being women - child care, elder parent care, family care, - in other words time available to do so.  So, even though I find nothing fundamentally wrong with this study, I don't think the results or the discussion of the results are/is discussing the real problem for PA in collegiate students in Qatar.  I would like the authors to discuss this topic further and offer us the data by gender on all tables.

Response: Thank you for this important remark.

  • We agree with the reviewer that the gender difference is the main result of our study.
  • To minimize selection bias, we used quota sampling method to recruit the students. The number of students per sex, nationality, and university of study is representative of the number of students for these categories in the Education City campuses. Therefore, the differences observed between study population groups are representative of the differences observed in the target population.
  • The difference in physical activity levels by sex has now been further analyzed as per the reviewer’s suggestion. Tables 1, 3, 4, 5 are now stratified by male and female.
  • We would like to highlight that our study focusses on university students at Education City campuses, Doha, Qatar. Barriers to physical activity such as childbearing and family care are likely faced by women in general and university students in other countries but less likely by female university students in Education City. In Qatar, official sources report that the female population in Qatar is marrying and building families at older age than previously observed, as their level of education is improving and they participate to the workforce (1). Between 2009 and 2015, average age at first marriage for Qatari females is 23.8 and for non-Qatari females is 26.0 (2). About 95% of our study population is undergraduate students.
  • Regarding the instruments used to measure the factors, our questionnaire was built using validated questionnaires (Pittsburgh Sleep Quality Index, Perceived Stress Scale 4 (PSS-4)) and questions from instruments in previously published primary studies. We have now further clarified this in the method section.

References

  1. Marriage and Family Doha, Qatar: Hukoomi, Qatar e-Government; 2021 [Available from: https://hukoomi.gov.qa/en/article/marriage-and-family.
  2. Marriage and Divorce. State of Qatar 2015 (Review and Analyasis) Doha, Qatar: Minister of Development Planning & Statistics; 2016 [Available from: https://www.psa.gov.qa/en/statistics/Statistical%20Releases/Population/MarriagesDivorces/2015/Analytical_Summary_Marriage_and_Divorce_2015_En.pdf.

Round 2

Reviewer 1 Report

I really appreciate the opportunnity to review this paper. I consider that all the changes have been included and now, the manuscript is ready to be published.